# Systemic Investigation Identifying Salivary miR-196b as a Promising Biomarker for Early Detection of Head-Neck Cancer and Oral Precancer Lesions

**DOI:** 10.3390/diagnostics11081411

**Published:** 2021-08-04

**Authors:** Ann-Joy Cheng, Guo-Rung You, Che-Jui Lee, Ya-Ching Lu, Shang-Ju Tang, Yi-Fang Huang, Yu-Chen Huang, Li-Yu Lee, Kang-Hsing Fan, Yen-Chao Chen, Shiang-Fu Huang, Joseph Tung-Chieh Chang

**Affiliations:** 1Department of Radiation Oncology and Proton Therapy Center, Linkou Chang Gung Memorial Hospital, Chang Gung University, Taoyuan 33302, Taiwan; annjoycheng@gap.cgu.edu.tw (A.-J.C.); kanghsing@adm.cgmh.org.tw (K.-H.F.); bigmac@adm.cgmh.org.tw (S.-F.H.); 2Department of Medical Biotechnology and Laboratory Science, College of Medicine, Chang Gung University, Taoyuan 33302, Taiwan; D000017007@cgu.edu.tw (G.-R.Y.); jerry.lee@kuleuven.be (C.-J.L.); D9601401@cgu.edu.tw (Y.-C.L.); D0701402@cgu.edu.tw (S.-J.T.); 3Graduate Institute of Biomedical Sciences, College of Medicine, Chang Gung University, Taoyuan 33302, Taiwan; 4Department of General Dentistry, Linkou Chang Gung Memorial Hospital, Taoyuan 33305, Taiwan; a7506@adm.cgmh.org.tw; 5School of Dentistry, College of Oral Medicine, Taipei Medical University, Taipei 11031, Taiwan; 6Graduate Institute of Dental and Craniofacial Science, College of Medicine, Chang Gung University, Taoyuan 33302, Taiwan; 7Department of Oral and Maxillofacial Surgery, Linkou Chang Gung Memorial Hospital, Taoyuan 33305, Taiwan; circlex@adm.cgmh.org.tw; 8Department of Pathology, Linkou Chang Gung Memorial Hospital, Taoyuan 33305, Taiwan; r22068@adm.cgmh.org.tw; 9Department of Radiation Oncology, New Taipei Municipal TuCheng Hospital, New Taipei City 236017, Taiwan; 10Department of Medical Imaging and Radiological Sciences, College of Medicine, Chang Gung University, Taoyuan 33302, Taiwan; 11Department of Radiation Oncology, Keelung Chang Gung Memorial Hospital, Keelung 20401, Taiwan; eric0705@adm.cgmh.org.tw; 12School of Medicine, Chang Gung University, Taoyuan 33302, Taiwan

**Keywords:** miR-196b, salivary biomarker, head-neck cancer, oral precancer lesion, diagnosis, early detection, miRNA panel

## Abstract

Background: Liquid biopsy is a rapidly growing field, for it may provide a minimally invasive way to acquire pathological data for personalized medicine. This study developed a systemic strategy to discover an effective salivary biomarker for early detection of patients with head-neck squamous carcinoma (HNSC) and oral precancer lesion (OPC). Methods: A total of 10 miRNAs were examined in parallel with multiple independent cohorts. These included a training set of salivary samples from HNSC patients, the TCGA-HNSC and GSE31277 cohorts to differentiate miRNAs between tumor and normal tissues, and groups of salivary samples from healthy individuals, patients with HNSC and OPC. Results: The combined results from the salivary training set and the TCGA-HNSC cohort showed that four miRNAs (miR-148b, miR-155, miR-196b, and miR-31) consistently increased in HNSC patients. Further integration with the GSE31277 cohort, two miRNAs (miR-31 and miR-196b) maintained at high significances. Further assessment showed that salivary miR-196b was a prominent diagnostic biomarker, as it remarkably discriminated between healthy individuals and patients with HNSC (*p* < 0.0001, AUC = 0.767, OR = 5.64) or OPC (*p* < 0.0001, AUC = 0.979, OR = 459). Conclusion: Salivary miR-196b could be an excellent biomarker for diagnosing OPC and early detection of HNSC. This molecule may be used for early screening high-risk groups of HNSC.

## 1. Introduction

Head-neck squamous cell carcinoma (HNSC) is one of the 10 leading cancers worldwide and is particularly prevalent in southeastern Asia [1,2,3]. The disease is more prevalent among males than females [1,2,3]. When HNSC occurs in the oral cavity region, it affects the patients’ life quality, especially in chewing, swallowing, and speech. Epidemiologic studies have shown a strong association between HNSC and environmental carcinogens, as habitual alcohol consumption, betel quid chewing, cigarette smoking, and HPV infection [3,4,5]. Upon carcinogen stimulation, HNSC may gradually develop through a multistep process in which genetic events disrupt the normal regulatory pathways. Oral premalignant lesions, including oral submucous fibrosis, dysplasia, and leukoplakia, have shown a high risk towards malignant transformation [3,4,5]. The survival rate of patients with early-stage HNSC is roughly 80%, but it is remarkably decreased by over 2-fold in the advanced-stage disease [3,6]. A clinically useful biomarker is required for early detection of HNSC to improve therapeutic efficacy.

At present, the histopathological examination or molecular analyses of the biopsies or resected tissues is the gold method for diagnosing HNSC. However, the biopsy procedure is very invasive and may not reflect the overall tumor heterogeneity [7]. Liquid biopsy has emerged as a minimally invasive and valuable tool to provide biomarker information for cancer management [8,9,10,11]. The scientific basis of liquid biopsy is that the diseased cells, such as a tumor, may release specific biomolecules into body fluids, these specific molecules thus represent a certain pathological status. For over a decade, the circulating nucleic acids have been considered valuable molecules of liquid biopsy, for they may be highly elevated in cancer patients [12,13]. Recently, microRNAs (miRNAs) have been found in various body fluids, including in plasma, saliva, and urine [14,15,16,17,18,19], providing a promising strategy to develop miRNAs as minimally invasive biomarkers.

MiRNAs are small (~17–22 nucleotides in length) non-coding RNAs that modulate gene expression by targeting mRNAs to induce translational repression [20,21]. Since miRNAs participate in many biological processes, including cell differentiation, proliferation, migration, and apoptosis, the dysregulation of these family molecules is associated with disease development [20,21]. Increasing studies have shown an association between the unique expression profile of miRNA and a specific type of cancer, including head-neck cancer [22,23,24,25,26,27,28,29,30]. Several miRNAs were commonly found in multiple lines of studies, including miR-21, miR-31, and miR-196b, indicating that these molecules possess high potential as useful cancer markers for clinical applications. Furthermore, cell-free miRNAs may be emergent candidates of circulating biomarkers for cancer diagnosis or prognosis. These molecules may be enclosed by secretory macrovesicle (such as exosomes) or complex with the Ago2 protein to escape endogenous ribonuclease degradation [31]. With the characteristics of miRNA in disease relevance and stable state in body fluid, miRNA is considered a novel class of clinically useful biomarkers.

In this study, we employ a systemic strategy to determine the potential use of salivary miRNAs in diagnosing HNSC. Five phases of studies were proceeded, such as the selection of 10 miRNA candidates; analyzing miRNAs in a training set of salivary samples from HNSC patients; examination of the differential miRNAs between tumor and normal tissues using the TCGA-HNSC cohort; verification of this panel miRNAs in pairs of cancer and normal tissues using an independent cohort (HNSC-GSE31277); and evaluation of salivary miRNA for the diagnosing efficacy in patients with HNSC and oral precancer lesions. We found that miR-196b was a prominent salivary biomarker for it can distinguish healthy individuals from patients with cancer or precancer lesions with high sensitivity and specificity.

## 2. Materials and Methods

### 2.1. Patients and the Saliva Sample Collection

This study was approved by the Institutional Review Board of Chang Gung Memorial Hospital, Taiwan. All of the patients and healthy individuals provided written informed consent indicating the willingness to donate their mouthwash for clinical research. The saliva samples were collected by oral swabbing from 86 patients with head-neck cancer and 30 patients with oral precancer lesions before the disease treatment. In addition, the saliva samples from 52 healthy individuals were also obtained for comparison. All of the oral swabbing samples were soaked in 1 mL of normal saline for over 30 min to release the miRNA molecules.

### 2.2. MicroRNA Extraction and Analysis

The miRNAs were enriched and purified from 200 μL saliva specimen and using the miRNeasy Mini Kit (Qiagen, Valencia, CA, USA), following the manufacturer’s protocol, and then eluted in 20 μL of nuclease-free water. The miRNA levels were determined by the miRCURY LNA™ Universal RT microRNA PCR system (Exiqon, Woburn, MA, USA). The reverse transcription reaction was performed by the addition of 12 μL of purified miRNA, 8 μL Universal cDNA synthesis reagent (Exiqon, Woburn, MA, USA) in a total reaction volume of 20 μL, followed by incubation at 42 °C for 60 min. A 1:50 dilution of RT product was used as the template for quantitative PCR. Quantitative PCR was performed in a similar manner, as previously described [32]. Briefly, the Bio-Rad CFX96 detection system (Bio-rad, Hercules, CA, USA) was carried out with a reaction mixture of 8 μL of the diluted reverse transcription reaction product, 2 μL of miRNA-specific PCR primer mix, and 10 μL of PCR master mix. The PCR results, recorded as quantitation cycle (Cq) values, were presented as relative fold expression. PCR reactions for quantifying miR-10b, miR-148b, miR-155, miR-184, miR-196b, miR-21, miR-31, miR-548b, miR-622, and miR-638 were performed in duplicate with SYBR^®®^ Green.

### 2.3. Bioinformatic Assessment of the Specific miRNAs in the Diagnostic Potential for HNSC

We performed the Oncomir Cancer Database (OMCD) online tool (https://www.oncomir.umn.edu/omcd/; accessed on 20 May 2021) to assess the diagnostic potential of the miRNAs in HNSC patients. This tool allows decoding an overall visualization of a miRNA profile from the TCGA-HNSC dataset and statistically analyzing the differentially expressed miRNAs between tumors and normal samples. The TCGA-HNSC dataset contains a miRNA profile from 532 miRseq data, including 488 tumor tissues with head-neck squamous cell carcinoma and 44 normal tissues. Additionally, an independent miRNA expression dataset GSE31277 was analyzed using GEO2R from the Gene Expression Omnibus (GEO) database of the NCBI. This dataset contains the data of Illumina miRNA microarrays from 15 pairs of tumor tissues and the normal tissues within cancer-free surgical margins from patients with oral squamous cell carcinoma. The significance of the differential expression level between tumor and normal tissues was examined by the *p*-value determined using the Student’s *t*-test.

### 2.4. Statistical Analyses

Mann–Whitney U analyses of variance were used to determine statistical differences of miRNA levels between unpaired groups (HNSC patients versus healthy individuals or tumor versus normal tissues). A *p*-value less than 0.05 was considered statistically significant. To evaluate the diagnostic efficacy of the salivary miR-196b, we analyzed the receiver operating characteristic (ROC) to differentiate two groups of samples (healthy individuals versus patients). We measured the area under the curve (AUC) to evaluate the discriminative power of salivary miR-196b. The clinical sensitivity and specificity for miR-196b were determined using Youden’s index analysis (sensitivity + specificity − 1 is maximal) to obtain an optimal cutoff threshold for maximum diagnostic effectiveness. Odds ratios were used to quantify how strong the salivary miR-196b may predict the risk with cancer or precancer. We performed the logistic regression analysis using SPSS 22 (IBM Corp., Armonk, NY, USA) or GraphPad Prism 9 (GraphPad Software Inc, San Diego, CA, USA).

## 3. Results

### 3.1. Analysis of 10 miRNAs from Saliva Specimen between Healthy Individuals and HNSC Patients

We developed a systemic strategy to discover an effective salivary miRNA biomarker for the detection of HNSC. Figure 1 showed the research design of this systemic study. Five phases of studies were employed such as the miRNA candidate selection; miRNA analysis; examination; verification; and evaluation. At the candidate selection phase, a total of 10 miRNAs were chosen for they were screened from our in-house findings or previously identified by several investigators [23,24,25,26,27,28,29,30]. These molecules were miR-10b, miR-148b, miR-155, miR-184, miR-196b, miR-21, miR-31, miR-548b, miR-622, and miR-638.

First, we established a miRNA detection protocol from saliva specimens and tested on a training set of samples. Oral swabbing was applied to obtain miRNA from the saliva. After extraction, the miRNAs were analyzed using a universal reverse transcription (RT)-miRNA PCR system (Exiqon). This RT protocol was used to determine multiple miRNAs with the same cDNA solution to minimize the potential variations caused by individual RT. Figure 2 showed the differential expression levels between 10 healthy individuals and 12 HNSC patients for each miRNA. As shown, all these miRNAs were overexpressed in HNSC patients on average. Except for miR-21, all these molecules exhibited statistical significance of overexpression (*p* < 0.05). These results suggested that our protocol of saliva miRNA detection was successfully established, and these miRNAs possessed high potential as saliva biomarkers to differentiate HNSC patients from healthy persons.

### 3.2. Examination of miRNA Biomarker Specific for HNSC Patients Using Two Independent Cohorts

Next, we applied the Oncomir Cancer Database (OMCD) online software to examine the distinguishing power of the 10 miRNAs using the TCGA-HNSC miRNA cohort. The TCGA-HNSC dataset contains a miRNA profile from 488 tumors and 44 normal tissues from HNSC patients. Figure 3A showed the overview of differential expression levels for these miRNAs between tumors and normal groups. Figure 3B showed the expression levels between tumors and normal groups of each miRNA. In this cohort, all miRNAs, except for miR-10b, exhibited consistent trends of overexpression in the cancer group in general. Note that the expression levels of miR-622 and miR-638 are much lower than the other miRNAs. These results may lead to a decrease in detection sensitivity and reduce the discrimination power. In these molecules, only five miRNAs exhibited statistical significance of over-expression in the cancer group. They were miR-148b (*p* < 0.0001), miR-155 (*p* = 0.0083), miR-196b (*p* < 0.0001), miR-21 (*p* < 0.0001), and miR-31 (*p* < 0.0001). These results provide an excellent panel of miRNA that may serve as diagnostic biomarkers to distinguish HNSC patients from healthy individuals.

To further identify the specific miRNAs that may participate in endogenous cancer progression, an independent cohort, GSE31277, which used tumors and the adjacent normal tissues from the same patients, was analyzed. This GSE31277 dataset comprised 15 pairs of the tumor and the surgical removed marginal normal tissues from HNSC patients. The Gene Expression Omnibus (GEO) online software was used to assess miRNA expressions. Figure 4A showed relative expressions of each miRNA between these two groups of tissues. In this cohort, most miRNAs showed no statistical significance (miR-148b, miR-155, miR-184, miR-21, miR-622, miR-638) or reduced expression (miR-10b and miR-548b) in the tumor tissues. These results indicated they played a less consequential function during oncogenic transformation. However, three miRNAs appeared at a significant level of upregulation in the tumor tissues (*p* < 0.0001 for miR-196b, *p* = 0.0412 for miR-31, and *p* = 0.0437 for miR-21), suggesting these molecules were critical in malignant formation.

Figure 4B showed the overall results of the 10 miRNA candidates assessed by three independent cohorts. The *p*-values presented the efficacy of each miRNA distinguishing between normal versus cancer groups. As shown, miR-196b and miR-31 consistently exhibited superior levels of overexpression in multiple study cohorts. Thus, these two miRNAs may be used for molecular signatures for further development of salivary diagnostic biomarkers.

### 3.3. Evaluation of Salivary miR-196b as a Diagnostic Biomarker for Detection of HNSC and PreMalignancy

In our defined panel candidates, miR-196b showed a domineering place of overexpression in three independent cohorts, this molecule was selected for further evaluation for diagnostic efficacy on HNSC and premalignancy. A total of 168 saliva samples from 52 normal individuals, 30 patients with oral precancer lesions (OPC), and 86 HNSC patients were obtained. The clinical characteristics of these participants were summarized in Table 1. The HNSC patients included 75 (87%) males and 11 (13%) females, with a mean age of 51 years. These OPC patients had similar gender and age characteristics with HNSC patients, including 26 (87%) males, 4 (13%) females, with a mean age of 50 years. Healthy individuals consisted of 36 (69%) males and 16 (31%) females, with a mean age of 45 years.

Figure 5A showed the relative differential levels of salivary miR-196b between healthy individuals and HNSC patients. As shown, the level of this molecule was significantly elevated in the cancer group, with an 11.2-fold increase on average (*p* < 0.001). The receiver operating characteristic (ROC) analysis was performed to evaluate the diagnostic performance of saliva miR-196b in distinguishing HNSC patients from healthy subjects. As shown, this molecule yielded an area under the curve (AUC) of 0.767. When applying Youden’s index (sensitivity + specificity − 1 is maximal) to obtain the optimal cutoff for diagnosis, the clinical sensitivity and specificity were 73% and 67%, respectively. To further acquire information regarding the risk level related to the elevation of miR-196b in saliva, the odds ratio analysis was determined, showing a 5.64 higher risk for HNSC (Table 2).

We further assessed the early detection power of salivary miR-196b by examining relative expression levels between healthy individuals and OPC patients. As shown in Figure 5B, miR-196b was substantially increased in the precancer group, with a 124-fold of elevation on average (*p* < 0.001). Using ROC analysis, this molecule yielded an AUC of 0.979 for the precancer patients. With optimal cutoff values, the sensitivity and specificity were 90% and 98% for diagnosing OPC patients. Additionally, we noted that miR-196b exhibited a greater diagnostic power in OPC patients compared to HNSC patients (Appendix A). This superior discrimination suggests that salivary miR-196b may be an excellent diagnostic biomarker for early detection of premalignancy to predict HNSC transformation.

## 4. Discussion

Liquid biopsy is a rapidly growing field in biomarker research, for it may provide a minimally invasive way to acquire physical pathological data for personalized medicine. Furthermore, early detection of HNSC is an optimal way for cancer prevention and treatment. In this regard, we developed a systemic strategy to discover an effective salivary biomarker to be developed for clinical practice (Figure 1). A total of 10 miRNAs were examined in parallel with multiple independent cohorts. As a result, we discovered that miR-196b was a prominent salivary biomarker for it can distinguish healthy individuals from patients with high sensitivity and specificity. Nevertheless, other miRNAs showing discrimination potentials in all cohort of studies may still be valuable, and the effectiveness to serve as salivary biomarkers is awaited for larger scaled validation.

In our training set of study, all the 10 miRNAs exhibited higher levels in HNSC patients compared to those from normal subjects (Figure 2). However, only five of them can be confirmed by the TCGA-HNSC dataset (Figure 3). Several factors can be addressed to explain this discrepancy aside from the difference in sample size. In sampling, our analysis in the training set used saliva specimens, while the TCGA cohort used dissected tissues or biopsy specimens. The salivary miRNA was thought to be protein-bound or enclosed by exosome-like vehicles and released into the circulation [31,32], whereas miRNAs from tissue samples represented the intracellular molecular condition. It is unclear whether miRNA may play a different pathogenic role in saliva or cells, whether the circulating miRNAs correlated with intracellular status in general or with a preferred selection to release a particular type of mRNA. All these questions are awaited to be investigated.

In the methodology, our analysis used a PCR-based method to examine a specific miRNA each time, while the TCGA acquired all molecular data simultaneously by the RNA sequencing method. For high-throughput screening, the simultaneous detection of numerous molecules is an effective approach. However, though this method is suitable to differentiate most of the molecules in a sample, it may be less sensitive to the very high-abundant molecules due to the detective saturation and overlook of the low-abundant molecules. In our study of 10-miRNA panel candidates, although miR-21 showed over-expression in the tumors from the data of the TCGA cohort, it has no statistical difference between healthy and cancer patients in our salivary training set of study. From the TCGA data, miR-21 is the highest abundant molecule, superiorly uppermost by all molecules. It is observed that although miR-21 was elevated in tumors compared to the normal tissues, the absolute level of this molecule in normal tissues still exhibited considerably higher than other oncomirs in tumors (Figure 3A, upper panel). It is assumed that the utmost level of cellular miR-21 may release a higher basal level of molecules into circulation, thus decreasing the sensitivity to differentiate between patients and healthy individuals. Therefore, the endogenous high level of miR-21 may account for reducing the discrimination power when applying salivary examination.

In the panel of 10-miRNA candidates, three molecules, miR-548b, miR-622, and miR-638, showed significantly higher levels in the HNSC patients of our training set (Figure 2) but no statistical difference in the tumors from the TCGA cohort (Figure 3). From TCGA data, it was observed that miR-548b, miR-622, and miR-638 exhibited very lower absolute levels across all of the samples, including in tumors or normal tissues (Figure 3A, upper panel). In general, when determining multiple molecules simultaneously in a sample, the molecules with low endogenous levels may lose their advantage as potential biomarkers due to slight detective sensitivity. Nevertheless, applying a single molecular assay to examine a specific biomarker may reduce the potential hindrance caused by abundant molecules. Thus, the discrimination of the relative level between two groups will appear. This situation may explain the difference of our results in the three low-abundant miRNAs (miR-548b, miR-622, and miR-638) inconsistent with the TCGA dataset. Thus, the discrimination potentials of these markers may still be valuable, and the effectiveness of being used as salivary biomarkers is awaited to be investigated.

In the panel of 10-miRNA candidates, four miRNAs (miR-148b, miR-155, miR-196b, and miR-31) showed a consistent trend in our training set and the TCGA cohort, as significant elevations in the HNSC patients (Figure 2) or the tumor tissues (Figure 3). These four molecules have been shown to overexpress and contribute to cancers by several lines of studies [23,24,25,26,27,28,29,30], suggesting the strong association of these molecules to HNSC. To assess the most prominent molecule that may be used as diagnostic biomarkers, we evaluated with an independent cohort (GSE31277) that consists of 15 pairs of tumor and normal tissues from the same patients. It is assumed that the elevation of tumor miRNAs in this cohort reveals a more direct and strong association of the specific molecules with malignant transformation. We found that only miR-31 and miR-196b increased expression in the tumors (Figure 4A), suggesting the crucial role of these two molecules in the tumorigenesis of HNSC. MiR-31 has played essential roles in regulating several malignant phenotypes, especially in cancer invasion and metastasis [33,34,35]. The salivary miR-31 was elevated in oral cancer and precancer patients, which may serve as a prognostic biomarker to predict cancer progression [36,37]. Similarly, miR-196 has been widely reported to play critical malignant roles in various cancers, this molecule was considered an emerging biomarker and therapeutic target [38,39,40]. In addition, the circulating miR-196b has also been examined as diagnostic or prognostic biomarkers in several cancers, including oral cancer [41,42,43]. These reports support our findings of the great prospective to apply miR-31 or miR-196b as HNSC biomarkers.

Since miR-196b showed a domineering level of overexpression in three independent cohorts, we evaluated the diagnostic efficacy of this molecule in HNSC. First, we assessed the differential levels of salivary miR-196b between healthy individuals and HNSC patients. We found that salivary miR-196b was significantly increased in the HNSC patients compared to healthy individuals (11.2-fold, *p* < 0.0001) (Figure 5A). The high values of AUC (0.767) and odds ratio (5.64) (Table 2) represent this molecule possessing a characteristic to be a useful diagnostic marker, as it effectively discriminates HNSC from the healthy group and substantially increases the risk of cancer incidence. The early diagnosis of cancer provides the opportunity to enhance its cure rate and improve the quality of life. In this regard, we further assessed the differential levels of salivary miR-196b between healthy individuals and patients with OPC. We found that this molecule even showed a greater diagnostic power by the remarkable increase in the precancer patients (124-fold, *p* < 0.0001) (Figure 5B). The exceedingly high levels of AUC (0.979) and odds ratio (459) (Table 2) provided clear segregation to distinguish OPC from the healthy group. The extremely high detection sensitivity and specificity (both >90%) were expected (Table 2). However, we noted that the diagnostic power of salivary miR-196b in OPC was superior to HNSC. The multistage theory of carcinogenesis has been developed to explain the transformation process from a normal cell into a cancer cell, as the initiation, promotion, progression, and finally malignancy. This micro-evolutionary process in each stage required the accumulation of a wide range of genetic alterations that affect cells to acquire new characteristics with growth advantage or possess other malignant properties [44,45,46]. In this study, we found that miR-196b was remarkably increased in the OPC patients. This result may reflect the molecular pathology that miR-196b plays a crucial role at the early tumor initiation stage. More apparently, we observed that most HNSC patients often lack saliva and may have difficulty obtaining proper swab sampling. This sampling problem may be the primary reason to explain why the diagnostic power is lower in the HNSC patients than in OPC. In the future, it may be better to apply mouth rinsing rather than oral swabbing to acquire saliva samples from HNSC patients. Still, the application of salivary specimens to determine miR-196b from OPC patients has shown a unique diagnostic approach.

## 5. Conclusions

This study demonstrated that salivary miR-196b could be an excellent biomarker for diagnosing OPC and early detection of HNSC. Our data serve as the knowledge foundation for further investigation by expanding into an extensive prospective study in the future. Salivary miR-196b may be incorporated into the routine clinical practice for early screening high-risk groups of HNSC.

## Figures and Tables

**Figure 1 diagnostics-11-01411-f001:**
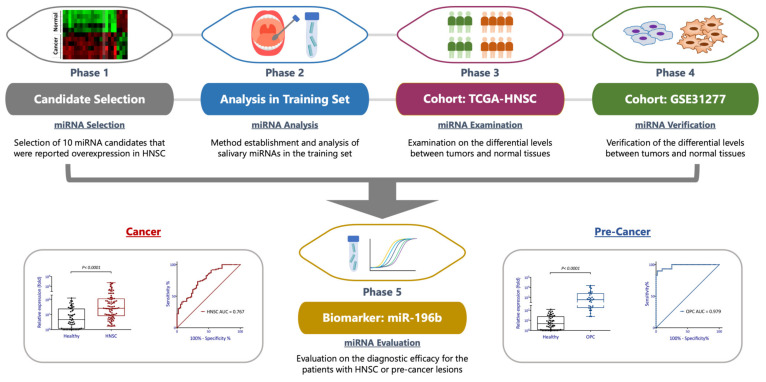
Flow diagram representing the systemic strategy and research design to discover an effective salivary biomarker for early detection of patients with head-neck cancer.

**Figure 2 diagnostics-11-01411-f002:**
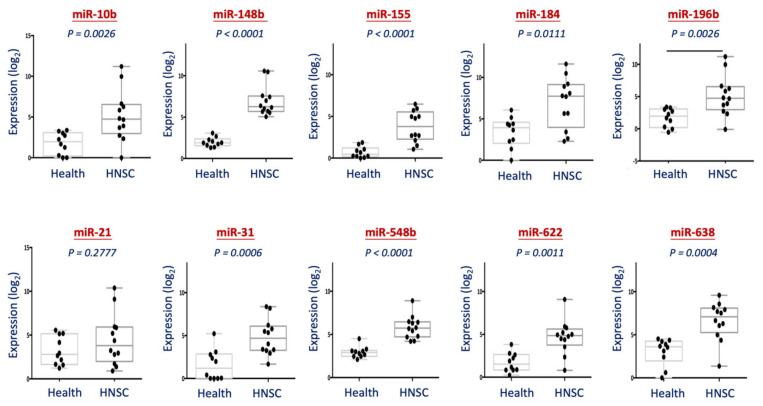
Analysis of the 10 miRNAs from saliva specimen between healthy individuals and HNSC patients. For each sample, the salivary miRNA was extracted and analyzed using a PCR-based method. The differential levels of the 10 miRNAs (miR-10b, miR-148b, miR-155, miR-184, miR-196b, miR-21, miR-31, miR-548b, miR-622, and miR-638) from healthy individuals and patients with HNSC were shown. The dot at the center of the box plot represents the mean expression for each miRNA.

**Figure 3 diagnostics-11-01411-f003:**
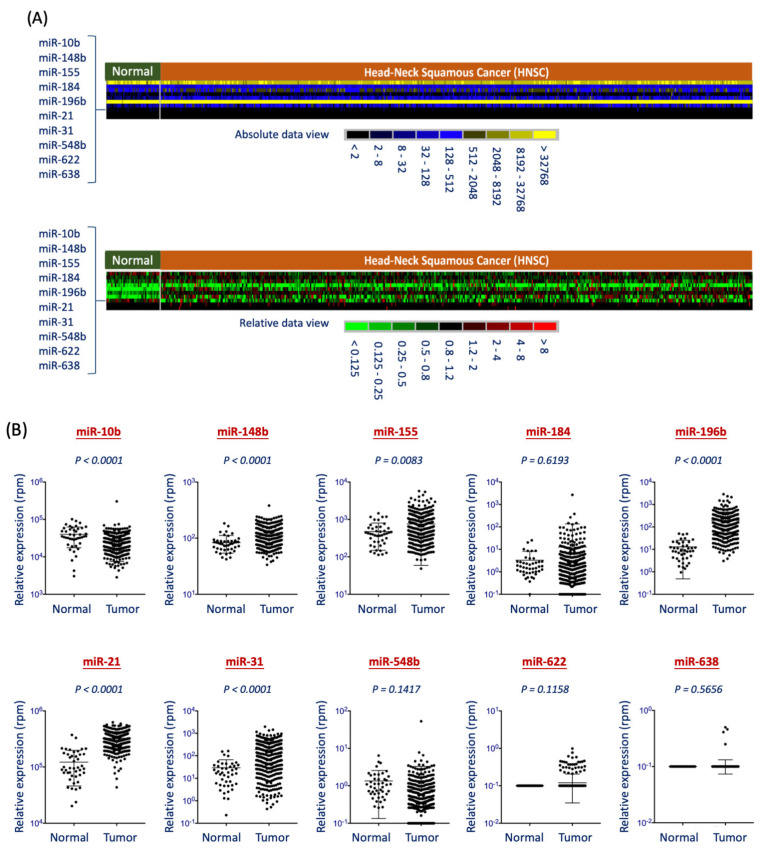
Examination of the 10 miRNA levels in the tumor and normal tissues from the TCGA-HNSC dataset. The Oncomir Cancer Database (OMCD) online software was used to examine the relative levels of the 10 miRNAs (miR-10b, miR-148b, miR-155, miR-184, miR-196b, miR-21, miR-31, miR-548b, miR-622, and miR-638) from the TCGA-HNSC miRNA cohort. This TCGA-HNSC dataset contains a miRNA profile from 488 tumors and 44 normal tissues from HNSC patients. (**A**) The overview of differential expression levels for these miRNAs between tumors and normal groups. Upper panel: The absolute levels across all miRNA molecules. Lower panel: The relative level in each miRNA molecule. (**B**) The expression levels between tumors and normal groups of each miRNA. The relative expression level with (rpm) was presented, derived from the read per million miRNA mapped (rpm) of the miRNAseq analysis used in the TCGA cohort.

**Figure 4 diagnostics-11-01411-f004:**
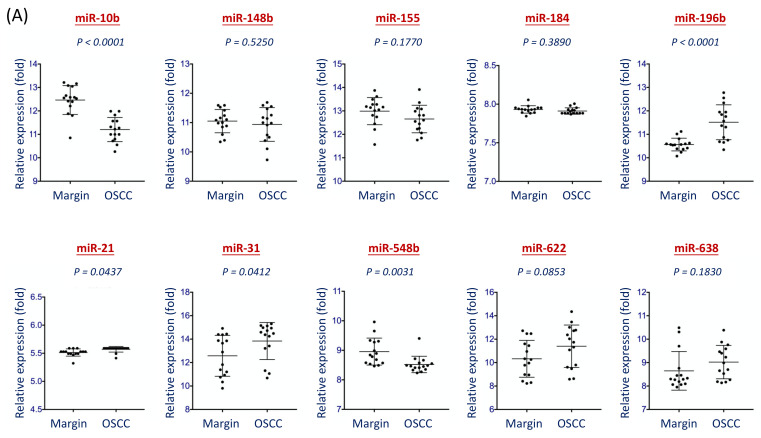
Examination of the 10 miRNA levels in the tumor and normal tissues from the GSE31277 dataset. The Gene Expression Omnibus (GEO) online software was used to determine the relative levels of the 10 miRNAs (miR-10b, miR-148b, miR-155, miR-184, miR-196b, miR-21, miR-31, miR-548b, miR-622, and miR-638) from the GSE31277 miRNA cohort. This GSE31277 dataset comprised 15 tumors and the surgical removed marginal normal tissues from HNSC patients. The relative expression level with (fold) was presented, after comparison with the signaling intensity of each data point in the Illumina miR array used in the GSE31277 cohort. (**A**) The expression levels between tumors and normal groups of each miRNA. (**B**) The overall results of the 10 miRNAs were assessed by three independent cohorts. The *p*-values presented the efficacy of each miRNA distinguishing between normal versus cancer groups.

**Figure 5 diagnostics-11-01411-f005:**
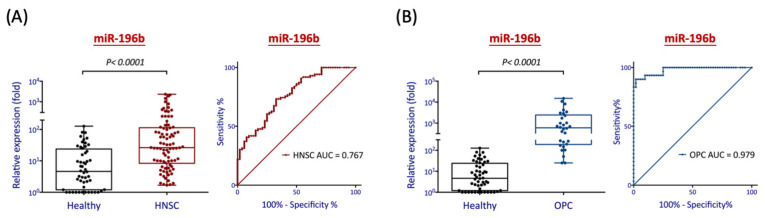
Evaluation of the diagnostic efficacy of miR-196b in the patients with HNSC or OPC. A total of 168 saliva samples from 52 normal individuals, 30 OPC patients, and 86 HNSC patients were recruited. (**A**) Relative expression levels and the significance of miR-196b in differentiation between healthy individuals and HNSC patients. (**B**) Relative expression levels and the significance of miR-196b in differentiation between healthy individuals and OPC patients. The relative expression level with (fold) was presented, after comparison to the level in OECM1 cancer cell lines (1:1000) by RT-qPCR analysis. The horizontal line across the center of the box plot represents the median value. Statistically significant differences were determined using the Mann–Whitney *U* test. The performance of receiver operating characteristic (ROC) curves and the area under the curve (AUC) of the ROC were shown.

**Table 1 diagnostics-11-01411-t001:** Clinical characteristics of the individuals that participated in this study.

Item	Cancer	Precancer	Normal
**Sex**			
Male	75 (87%)	26 (87%)	36 (69%)
Female	11 (13%)	4 (13%)	16 (31%)
**Age**			
Range (year old)	19–82	33–81	22–73
Mean ± SD (years)	50.8 ± 12.4	49.5 ± 13.4	45.3 ± 11.2
<40 years old	14 (16%)	9 (30%)	12 (23%)
40–49 years old	27 (31%)	7 (23%)	18 (35%)
50–59 years old	24 (28%)	7 (23%)	17 (33%)
≥60 years old	21 (24%)	7 (23%)	5 (10%)
**Habits**			
Alcohol drinking	42 (48%)	12 (40%)	
Betel quid chewing	40 (47%)	9 (30%)	
Cigarette smoking	49 (57%)	20 (67%)	
**Pathologic T-status**			
T1-T2	41 (48%)		
T3-T4	45 (52%)		
**Pathologic N-status**			
pN0	28 (33%)		
pN+	58 (67%)		
**Pathologic overall stage**			
I–II	21 (24%)		
III–IV	65 (76%)		
Total	86 (100%)	30 (100%)	52 (100%)

**Table 2 diagnostics-11-01411-t002:** Diagnostic efficacy for saliva miR-196b levels in patients with HNSC and oral precancer.

Saliva miR-196b					
Disease	*p*-Value	AUC	Sensitivity	Specificity	Odds Ratio
			(% Range)	(% Range)	(Range)
**HNSC**	<0.001	0.767	73.3%	67.3 %	5.64
			(63~83)	(53~80)	(2.7~12.0)
**Oral Precancer**	<0.001	0.979	90.0%	98.1%	459
			(74~98)	(90~100)	(45~4627)

## Data Availability

The data presented in this study are available on request from the corresponding author.

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
