# Peer review of "Systemic Investigation Identifying Salivary miR-196b as a Promising Biomarker for Early Detection of Head-Neck Cancer and Oral Precancer Lesions"

_diagnostics, 2021, doi:10.3390/diagnostics11081411_

Round 1
Reviewer 1 Report
Additional graphical representation for miR-196b for healthy control, OPC and HNSC.
Please add the number of patients used for miRNA evaluation as validation step.
Please add line of identity for ROC curves.
Author Response
Dear Reviewers and Editor:
We thank the comments for helping to make this manuscript more complete. All
the comments have been well taken and amended in the revised manuscript with
“Track Changes” function. We have made a point-by-point response to the reviewers’
comments as below. Thank you very much!
Sincerely,
Joseph T Chang, MD, MPH
Chang Gung Memorial Hospital and Chang Gung University, Taiwan
Response to Reviewer 1 Comments
General comments
1: Additional graphical representation for miR-196b for healthy control, OPC, and
HNSC.
Response: Thanks for the comment. The graphical representation of miR-196b for the
healthy group, OPC and HNSC were shown in Supplemental Figure S1. “Additionally,
we noted that miR-196b exhibited a greater diagnostic power in OPC patients
compared to HNSC patients (Figure S1). (Lines 281-283).
2: Please add the number of patients used for miRNA evaluation as validation step.
Response: The patient number used in the validation step was described in Method
section 2.1 and Figure 5 legend. “A total of 168 saliva samples from 52 normal
individuals, 30 OPC patients, and 86 HNSC patients were recruited”. (Lines 319-320)
3: Please add the line of identity for ROC curves.
Response: Thanks for the comment. We have added central lines in the ROC curves of
the revised Figures 5A and 5B.

Reviewer 2 Report
This study uses liquid biopsy together with public microRNA datasets to examine the diagnosis roles of candidate microRNAs in head and neck cancer and identifies salivary miR-196b as a biomarker for head and neck cancer and oral pre-cancer lesions. The early diagnosis roles of circulating miR-196b and miR-10b in oral cancer or pre-cancer lesions have been reported previously by this group (doi: 10.1158/1940-6207.CAPR-11-0358; 10.1016/j.clinbiochem.2014.11.020). Thus, the novelty of the present study is not high.
Major:
Fig 2, 4B, the training set contains only 12 tumor and 10 normal salivary samples, which may have a concern on sampling bias and thus some results may be not representative. This issue should be considered especially the authors find a difficulty in the sampling of cancer cases (lines 405-406). This reviewer wonders whether the results of miR-21 (and miR-10b and some others) will be different if the authors test it using their entire cohort (86 tumor, 52 normal, and 30 pre-cancer samples).
Minor:
Fig 3B, 4A, 5, please define the expression unit in Y-axis. (fold? Comparison of…?)
TCGA_HNSC microRNA dataset should contains >522, not 488, cancer samples, among which 44 have adjacent normal parts. Please confirm.
Fig 5 legend, “violin plot” should be “box plot”, “medium” should be “median” (line 313).
Line 404, please explain more in detail for the tumor initiation role of miR-196b.
Please confirm the unit: “200 mL” saliva specimen (line 116) and “20 mL” of nuclease-free water (line 118).
Author Response
Dear Reviewers and Editor:
We thank the comments for helping to make this manuscript more complete. All
the comments have been well taken and amended in the revised manuscript with
“Track Changes” function. We have made a point-by-point response to the reviewers’
comments as below. Thank you very much!
Sincerely,
Joseph T Chang, MD, MPH
Chang Gung Memorial Hospital and Chang Gung University, Taiwan
Response to Reviewer 2 Comments
General Comments:
This study uses liquid biopsy together with public microRNA datasets to examine the
diagnosis roles of candidate microRNAs in head and neck cancer and identifies salivary
miR-196b as a biomarker for head and neck cancer and oral pre-cancer lesions. The
early diagnosis roles of circulating miR-196b and miR-10b in oral cancer or pre-cancer
lesions have been reported previously by this group (doi: 10.1158/1940-6207.CAPR-
11-0358; 10.1016/j.clinbiochem.2014.11.020). Thus, the novelty of the present study
is not high.
Response: Thanks for the comment. Previously, we have found that miR-196b is
elevated in the plasma of head-neck cancer patients, and this molecule playing a
significant role in regulating several malignant phenotypes. This work extends our
previous studies to examine whether miRNA can serve as a salivary biomarker for
screening or diagnosing head-neck cancer. We also evaluated 10 miRNA molecules in
parallel to assess their potentials in diagnostic efficacy. We conclude that salivary miR-
196b could be an excellent biomarker for diagnosing oral pre-cancer and early
detection of head-neck cancer. In this work, the research design is novel, and our
finding is significant. Our conclusion in this study is worthy of being noted in the field
of translation oncology.
Major question:
Fig 2, 4B, the training set contains only 12 tumor and 10 normal salivary samples,
which may have a concern on sampling bias and thus some results may be not
representative. This issue should be considered especially the authors find a difficulty
in the sampling of cancer cases (lines 405-406). This reviewer wonders whether the
results of miR-21 (and miR-10b and some others) will be different if the authors test it
using their entire cohort (86 tumor, 52 normal, and 30 pre-cancer samples).
Response: Thanks for the question. In the training set of study (first cohort), we used
saliva samples of 10 normal individuals and 12 cancer patients to examine 10 miRNA
molecules in parallel. Several molecules showed promises to differentiate between
normal and cancer groups. Further integrated analyses with TCGA-HNSC and
GSE31277 datasets, miR-196b showed a domineering place to distinguish these two
groups in three independent cohorts. We observed that the endogenous highabundant
(such as miR-21) or low-abundant (such as miR-638) molecules might lose
their advantage as biomarkers due to the reduction of differential specificity or
sensitivity. We further selected miR-196b for confirmational evaluation and
demonstrated it as an excellent biomarker to distinguish normal from cancer/precancer.
However, we agreed with the reviewer’s comment that other miRNA molecules,
such as miR-21, may be a promising biomarker if we expand our study to the entire
conformational cohort. Without rule out this possibility, we add a note in the
discussion section. “Nevertheless, other miRNAs showing discrimination potentials in
all cohort of studies may still be valuable, and the effectiveness to serve as salivary
biomarkers is awaited for larger scaled validation.” (Lines 336-338).
Minor questions:
1. Fig 3B, 4A, 5, please define the expression unit in Y-axis. (fold? Comparison
of…?)
Response: For these figures, the expression units in Y-axis were added and
described in the figure legend of the revised manuscript.
In Figure 3B. “The relative expression level with (rpm) was presented, derived
from the read per million miRNA mapped (rpm) of the miRNAseq analysis used
in the TCGA cohort.” (Line 220-221)
Figure 4A. “The relative expression level with (fold) was presented, after
comparison with the signaling intensity of each data point in the Illumina miR
array used in the GSE31277 cohort.” (Line 249-251)
Figure 5. “The relative expression level with (fold) was presented, after
comparison to the level in OECM1 cancer cell lines (1:1000) by RT-qPCR
analysis.”(Line 322 - 323)
2. TCGA_HNSC microRNA dataset should contains >522, not 488, cancer samples,
among which 44 have adjacent normal parts. Please confirm.
Response: Thanks for the question. The TCGA-HNSC contained 528 cases,
However, in the Oncomir Cancer Dataset (OMCD), only 488 cancer samples have
microRNA data, while all the 44 normal samples have microRNA data.
3. Fig 5 legend, “violin plot” should be “box plot”, “medium” should be “median”
(line 313).
Response: Thanks for the correction. In Figure 5 legend, the “violine plot” has
changed to the “box plot”, and the “medium” changed to “median”.
4. Line 404, please explain more in detail for the tumor initiation role of miR-196b.
Response: We have added the information to explain the potential role of miR-
196b in the tumor initiation stage. “The multistage theory of carcinogenesis has
been developed to explain the transformation process from a normal cell into a
cancer cell, as the initiation, promotion, progression, and finally malignancy. This
micro-evolutionary process in each stage required the accumulation of a wide
range of genetic alterations that affecting cells to acquire new characteristics with
growth advantage or possessing other malignant properties [44-46]. In this study,
we found that miR-196b was remarkable increased in the OPC patients.” (Lines
419-425). We have cited references in the revised manuscript.
5. Please confirm the unit: “200 mL” saliva specimen (line 116) and “20 mL” of
nuclease-free water (line 118).
Response: Thanks for the correction. The typo has been changed.

Round 2
Reviewer 2 Report
The revised manuscript has been improved and is acceptable for publication.